# Therapeutic Drug Monitoring as a Tool for the Clinical Outcome Prediction in Vedolizumab-Treated Patients: An Italian Pilot Study

**DOI:** 10.3390/biomedicines12040824

**Published:** 2024-04-09

**Authors:** Jessica Cusato, Davide Giuseppe Ribaldone, Michela Helga Falzone, Alessandra Manca, Miriam Antonucci, Alice Palermiti, Giorgio Maria Saracco, Linda Ceccarelli, Francesco Costa, Andrea Bottari, Ginevra Fornaroli, Gian Paolo Caviglia, Antonio D’Avolio, Lorenzo Bertani

**Affiliations:** 1Laboratory of Clinical Pharmacology and Pharmacogenetics, Department of Medical Sciences, Amedeo di Savoia Hospital, University of Turin, Corso Svizzera, 164, 10149 Turin, Italy; jessica.cusato@unito.it (J.C.); alice.palermiti@unito.it (A.P.); 2Unit of Gastroenterology, Department of Medical Sciences, University of Turin, 10124 Turin, Italy; davidegiuseppe.ribaldone@unito.it (D.G.R.); helga.f.98@me.com (M.H.F.); giorgiomaria.saracco@unito.it (G.M.S.); 3SCDU Infectious Diseases, Amedeo di Savoia Hospital, ASL Città di Torino, 10149 Turin, Italy; miriam.antonucc@aslcittaditorino.it (M.A.); gianpaolo.caviglia@unito.it (G.P.C.); 4IBD Unit, Department of General Surgery and Gastroenterology, Pisa University Hospital, 56124 Pisa, Italy; ceccarellilinda@gmail.com (L.C.); fcosta@med.unipi.it (F.C.); lorenzo.bertani@gmail.com (L.B.); 5Gastroenterology Unit, University of Pisa, 56126 Pisa, Italy; a.bottari@studenti.unipi.it (A.B.); ginervra.fornaroli@gmail.com (G.F.)

**Keywords:** IBD, TDM, monoclonal antibody, personalized therapy

## Abstract

Over the years, vedolizumab (VDZ) has emerged as a more effective target therapy for inflammatory bowel disease. The aim of this work was to analyze a cohort of inflammatory bowel disease patients, evaluating the association between VDZ serum concentrations at 6 months from starting therapy and their clinical and biochemical indexes within one year of treatment, correlating drug levels with response and clinical remission. Forty patients treated with VDZ were enrolled. Drug concentrations were quantified through ELISA methods. VDZ levels correlated with hemoglobin levels at twelve months of therapy (*p* = 0.03) and with clinical remission at twelve months of therapy (*p* = 0.03); patients who reached clinical remission showed higher VDZ concentrations. A VDZ cut-off value of 43.1 μg/mL was suggested, predicting clinical remission at twelve months of therapy. A statistically significant association between VDZ levels at T6 and calprotectin <250 μg/g at T12 was found (*p* = 0.04). Furthermore, the optimal threshold value of VDZ levels at T6 associated with calprotectin <250 μg/g at T12 was identified: through levels higher than 45.2 µg/mL, we were able to predict remission one year after therapy. In the final regression multivariate model, no factor was retained as a predictor of clinical remission at one year of treatment. In conclusion, this is the first pilot study reporting a possible VDZ serum cut-off value able to predict not only the clinical remission at twelve months of therapy but also the calprotectin level, which is very important, as it is a surrogate marker of mucosal healing.

## 1. Introduction

Inflammatory bowel diseases include Crohn’s disease and ulcerative colitis. These pathologies are increasing worldwide, and they are multifactorial, lifelong, inflammatory diseases of the gastrointestinal tract [1]. Unfortunately, there is not a definitive cure, but several new drugs (including biologics and small molecules) have been approved in the last few years, showing an improvement in terms of course and the quality of life of patients with these pathology [2,3,4].

In particular, in the last few years, the inflammatory bowel disease burden is rising globally, with differences in terms of levels and trends. It is important to understand these geographical differences in order to act effective strategies for preventing and treating these pathologies. For example, some authors analyzed the prevalence, mortality, and overall burden of inflammatory bowel diseases in 195 different countries and territories between 1990 and 2017, focusing on data from the Global Burden of Diseases, Injuries, and Risk Factors Study (GBD) 2017: they reported that in 2017, 6.8 million cases of inflammatory bowel diseases were present globally. The age-standardized prevalence rate increased from 79.5 per 100,000 people in 1990 to 84.3 per 100,000 people in 2017, whereas the age-standardized death rate decreased from 0.61 per 100,000 people in 1990 to 0.51 per 100,000 people in 2017. In addition, they reported that the highest age-standardized prevalence rate in 2017 was present in high-income North America (422.0 per 100,000) and the lowest age-standardized prevalence rates in the Caribbean (6.7 per 100,000 people). Considering the national level, the USA had the highest age-standardized prevalence rate, followed by the UK. Vanuatu had the highest age-standardized death rate in 2017, while Singapore had the lowest. The authors concluded by suggesting these pathologies are an important social and economic burden on governments and health systems to be considered in the future. In fact, they highlight that their findings can be useful for policy makers developing strategies to manage inflammatory bowel diseases, also considering the education of specialized health personnel to address the burden of this complex disease [5,6,7].

It is important to highlight that, unfortunately, the clinical remission outcome rate reached with these drugs is only about 30%, with a significant increase in costs and side effects. These could be in part avoided if therapy could be personalized, identifying the right drug for the right patient at the right dosage, as reported in the literature [8]. In this perspective, many different biomarkers have been suggested in clinical practice in order to predict the therapeutic efficacy [9,10,11], but their use is still limited since the approach with a single predictor highlights some concerns [12]. Consequently, the future of inflammatory bowel disease research is changing through the recognition of the greatest number of possible biomarkers, with the aim of developing a possible model integrating serum or stool biomarkers [13]. 

Vedolizumab (VDZ) is a gut-selective anti-inflammatory monoclonal antibody, which selectively binds to the α 4 β 7 integrin and blocks its interaction with mucosal addressin cell adhesion molecule-1 (MAdCAM-1) [14].

VDZ was approved by the European Medicines Agency (EMA) and Food and Drug Administration (FDA) for the inflammatory bowel diseases, particularly ulcerative colitis and Crohn’s disease treatment, in patients who have an inadequate response to standard therapies [14]. This drug is used as a second-line strategy to achieve clinical remission and, possibly, also endoscopic remission in moderate-to-severe active inflammatory bowel disease and as maintenance therapy. Clinical remission is obtained in about 40% of patients, and among those who initially benefit from it, every year a rate of the loss of response to therapy of around 10–15% is described [15]. In addition, it is an expensive drug. Consequently, it becomes essential to identify predictors of clinical response in order to improve the cost–benefit ratio of the treatment.

Considering pharmacokinetic characteristics, VDZ shows a slow linear elimination until approximately 10 μg/mL, but the elimination process is faster and non-linear at reduced levels [16]. VDZ has a prolonged half-life of about 25.5 days during the linear elimination. Clearance is higher in people with severe obesity (>120 kg) and low albumin levels (<3.2 g/dL). The VDZ half-life and clearance do not change at doses >2.0 μg/mL, and the increase in exposure is proportional to the VDZ dosage. Elimination occurs through cellular uptake and consequent proteolytic degradation, and the clearance is regulated by receptors [16]. 

As for other monoclonal antibodies, increased inflammation and neutralizing antibody production unfortunately lead to a higher VDZ clearance [17]. In fact, although the anti-VDZ antibody presence is basically low (6%), a reduced VDZ exposure, with a consequent reduction in treatment efficacy, was suggested. Moreover, efficacy was associated with increased VDZ concentrations. In this context, it is very important to highlight that the role of therapeutic drug monitoring (TDM) in predicting the clinical outcome for VDZ treatment is still unclear: although some evidences have been reported in the last few years, a few data in real-life are available to guide clinicians on the optimal dosing [16]. TDM is the clinical practice helpful in quantifying drug concentrations in the patient bloodstream, aiming at optimizing individual dosage regimens. Particularly, it is performed for drugs with narrow therapeutic ranges, drugs with increased pharmacokinetic variability, drugs with several and severe adverse reactions and substances with target levels difficult to monitor. Consequently, it allows for the use of difficult-to-manage drugs’ appropriate exposure and to optimize the clinical outcome in terms of efficacy or tolerability [18]. Different studies suggest the importance of performing TDM: as an example, there are potential concerns about antiretroviral drug efficacy and/or tolerability in particular patients, who could, probably, obtain an improvement from the application of TDM in the HIV field [19]. In fact, there are some clinical settings in which the recognition of the optimal antiretroviral treatment is a challenge, for example, wide polypharmacy resulting in an increased probability of drug–drug interactions. In addition, TDM is important in the context of specific populations, such as elderly or pediatric patients, gastrectomized patients, or pregnant women. Furthermore, TDM could have an impact in patients with a known or suspected history of scarce adherence to therapies and/or the appropriate dosage regimen or patients with resistance to some HIV antiretroviral agents [20]. 

Concerning the TDM of anti-TNF treatment, in the recent years, it revolutionized the modern management of inflammatory bowel diseases. Indeed, up to 30% of patients were non-responders, and in responder patients, the subsequent secondary loss of response is highlighted in up to 30% after one year of treatment and about 20% annually thereafter [21]. 

TDM is a useful clinical practice for tailoring treatment in some clinical contexts, including inflammatory bowel diseases [18,22]. It could be reactive in order to identify the cause of a loss of response to a therapy or proactive, aimed at steadily maintaining the drug concentrations in the therapeutic range, in order to prevent the loss of response. For monoclonal antibodies, drug and anti-antibody concentrations are quantified in order to maximize efficacy and reduce side effects. Due to immunogenicity, antidrug antibodies can develop; this could lead to augmented drug clearance and consequent reduced drug exposure, with a reduced probability of optimal clinical response.

Several studies showed that infliximab or adalimumab levels are increased in responders compared with non-responders. In fact, some trough level-based therapy algorithms have been proposed to guide the clinicians in the treatment choice [23,24,25]. Concerning VDZ, no TDM recommendations are proposed currently. In this context, some studies have been published: Levartovsky et al. analyzed 86 inflammatory bowel disease patients (51 patients with Crohn’s disease and 35 with ulcerative colitis) who discontinued VDZ. In this study, 72 (83.7%) stopped VDZ therapy because of a loss of response, but their trough levels at discontinuation were not different compared to patients with a clinical response. It is important to highlight that patients progressing to subsequent surgery had reduced VDZ concentrations compared with patients who were treated with an additional medical therapy [26]. 

A study of the University of Michigan analyzed 472 patients with Crohn’s disease from the VDZ registration trials, performing random forest machine learning algorithm modeling on first testing and then validation cohorts to suggest early predictors of remission in the absence of corticosteroids at week 52. Data and factors from the baseline to week 6, including VDZ trough levels, were used to build this study model: week 6 serum VDZ levels were one of the five strongest predictors of remission in the model [27]. 

In another post hoc analysis of the GEMINI 1 study, early VDZ trough levels at weeks 2, 4, and 6 were associated with clinical remission at week 14. These patients showed higher median VDZ levels at 2, 4, and 6 weeks compared with those with active pathology. When stratified into quartiles by trough concentration only, higher trough levels at week 6 were associated with an increased remission outcome at week 14, without identifying possible cut-off values [27]. 

In a cross-sectional study of Al-Bawardy et al., 171 patients with a diagnosis of IBD and treated with VDZ were enrolled in order to determine the median VDZ trough levels, their correlation with CRP, mucosal healing (absence of mucosal ulcers in Crohn’s disease), and the change in clinical management. The median VDZ was 15.3 µg/mL (range, 0–60). Patients with a normal CRP showed median VDZ levels higher than patients with high CRP: 17.3 µg/mL vs. 10.7 µg/mL, respectively (*p* = 0.046). Statistically significant differences were found in patients with Crohn’s disease (*p* = 0.005) but not in ulcerative colitis ones (*p* = 0.72). Mucosal healing was achieved in 35% of patients: in these subjects, median VDZ was 13.7 µg/mL, while it was 16.1 µg/mL (*p* = 0.64) in patients who did not achieve mucosal healing. In conclusion, VDZ levels resulted in an impact on the clinical management in 73% of treated subjects [28]. 

Another study [29] aims to explain the predictive role of VDZ levels in long-term clinical outcomes in treated patients: 95 subjects were included. An amount of 29.5% of patients with a mean VDZ treatment time of 17.83 months ± 14.56 showed a clinical response, while 45.3% were in clinical remission at the end of the study. The VDZ mean level at week 6 was 41.79 µg/mL ± 24.58. Association between VDZ levels at week 6 and both short- and long-term outcomes could not be demonstrated. Nevertheless, higher VDZ levels were observed in subjects with endoscopic and clinical improvement at month 6 and at the time of the last follow-up [29]. 

In addition, in the study of Plevris et al., the relationship between C_trough_ VDZ levels and outcomes during maintenance treatment was investigated. Seventy-three patients were enrolled, and VDZ levels were matched with clinical activity scores, CRP, and fecal calprotectin, and only forty patients were also matched with endoscopic data. Similar median VDZ levels were observed in both patients in clinical remission and not in clinical remission: 10.6 and 9.9 µg/mL, respectively (*p* = 0.54); this was also shown for biologic remission (10.6 vs. 9.8 µg/mL, *p* = 0.35) and endoscopic remission (8.1 vs. 10.2 µg/mL, *p* = 0.21). No significant increase in subjects in clinical remission, biologic remission, or endoscopic remission with increasing C_trough_ VDZ levels (*p* < 0.05) was observed [30].

Improving the timing of response to anti-TNF therapy is therefore critical, considering the relative lack of alternative effective therapy options and the increased burden of costs. Different data suggest a clear relationship between exposure and response concerning anti-TNF therapy; consequently, minimum circulating drug level cut-offs are related to an improved clinical outcome. Considering Crohn’s disease, for example, infliximab drug exposure higher than 5 μg/mL and adalimumab exposure higher than 4.95 μg/mL have been shown to be associated with clinical remission in cross-sectional studies [21]. 

Considering all these data and since few results are present in the literature concerning VDZ TDM, the aim of this study was to evaluate the VDZ drug concentration influence in predicting the clinical outcome in a cohort of Italian patients with inflammatory bowel disease, considering the clinical response at six and twelve months of therapy.

## 2. Materials and Methods

### 2.1. Study Design

A prospective, two-center study at the Gastroenterology Unit of “Città della Salute e della Scienza di Torino”, Italy, and the Inflammatory Bowel Diseases Unit, Pisa University Hospital, Pisa, Italy, was performed. From July 2016 to July 2022, patients treated with VDZ were enrolled (some concerns were related to COVID-19 pandemic). All patients signed the informed consent to participate in the study. 

Inclusion criteria were patients treated with VDZ for both ulcerative colitis and Crohn’s disease for at least six months of therapy in order to obtain a blood withdrawal before the next drug administration (C_trough_). All patients were treated with intravenous VDZ. Every patient received 300 mg of VDZ intravenously at baseline, after 2 weeks, and after 6 weeks (induction phase), then every 8 weeks (maintenance phase). Patients receiving VDZ every 4 weeks in the first 6 months of the follow-up were excluded from the study. VDZ was quantified in serum after 6 months of therapy (T6), immediately before an administration (trough levels). Consequently, C_trough_ were basically obtained at week twenty-two (5th administration). Exclusion criteria were patients stopping VDZ before six months of therapy because they were not in a stable condition of drug concentration. No subcutaneous administrations were considered. 

Clinical characteristics were evaluated at three different time-points: before the beginning of VDZ administration (T0), after 6 months of therapy (T6), and at 12 months of therapy (T12). At the same time-points, C-reactive protein (CRP), fecal calprotectin, and hemoglobin (Hb) values were assessed.

The outcomes included the prediction of corticosteroid-free clinical response at 6 and 12 months of VDZ therapy (T6 and T12). Clinical response to VDZ therapy was defined in accordance with the International Organization for the Study of Inflammatory Bowel Disease (IOIBD) [31]:-For Crohn’s disease, at least a 3-point decrease in Harvey–Bradshaw Index (HBI) from before treatment initiation or HBI ≤ 4 at the time of assessment;-For ulcerative colitis, a decrease of at least 2 points in partial MAYO score (PMS) or PMS ≤ 1.

Predictors of clinical remission, defined as HBI ≤ 4 for Crohn’s disease and PMS ≤ 1 for ulcerative colitis, were also checked.

VDZ through levels at T6 were also correlated with calprotectin <250 μg/g at T12.

Stopping VDZ was considered a failure to reach the outcomes.

The study followed the principles of the Declaration of Helsinki and was approved by the local ethical committees: Comitato Etico Interaziendale A.O.U. Città della Salute e della Scienza di Torino—A.O. Ordine Mauriziano—A.S.L. Città di Torino (approval code n. 0056924); Comitato Etico Regionale Toscana Area Vasta Nord Ovest—CEAVNO (approval code n. 16790).

### 2.2. Vedolizumab and Antibody–Anti Vedolizumab Quantification

Blood collection was taken before the new dose administration (C_trough_). 

Serum samples were isolated after whole blood centrifugation at 1400× *g* for 10 min at 4 °C. Samples were stored at −80 °C until the analysis. 

VDZ serum concentrations were obtained with ELISA (enzyme-linked immunosorbent assay) technique using SHIKARI ^®^ kit (Q-VEDO, MATRIKS BIOTECHNOLOGY CO., LDT., Ankara, Turkey), while Ab anti-VDZ serum concentrations were determined through SHIKARI^®^ (S-ATV) kit (Q-VEDO, MATRIKS BIOTECHNOLOGY CO., LDT., Ankara, Turkey). 

### 2.3. Statistical Analysis

All variables were tested for normality through the Shapiro–Wilk test. For normally distributed variables, mean and standard deviation (SD) were reported, whereas for non-normally distributed variables, median and interquartile range (IQR) were reported. Categorical variables were reported as numbers and percentages. Continuous variables were correlated with serum VDZ concentrations, yielding a correlation coefficient (r); instead, categorical variables were tested with Student’s *t*-test for independent samples.

The association between the variables was evaluated using logistic regression; the statistically significant variables of the univariate analysis (enter regression) were included in the multivariate analysis. 

ROC curve analysis and Youden index were used to calculate threshold serum VDZ levels at T6 to predict clinical remission at T12. 

Statistical analysis was performed with MedCalc^®^ Statistical Software version 20.104, Med Calc Software Ltd., Ostend, Belgium; https://www.medcalc.org; 2022.

## 3. Results

Forty patients were recruited from “AOU Città della Salute e della Scienza” from Turin and “AOU Pisa” Hospitals.

Patient characteristics and therapy at the first timepoint (T0) are reported in Table 1. In this study, 16 subjects (40%) were affected by Crohn’s disease, while 24 (60%) were affected by Ulcerative colitis. 

At each time point, clinical and hematochemical data were collected and reported in Table 2.

After 6 months of treatment, mean VDZ trough levels were 79.3 µg/mL (SD = 51.3), and no anti-drug antibodies were detected.

Corticosteroid-free clinical response and remission rate at T6 and T12 are reported in Table 3.

At T6, correlations between VDZ trough levels and HBI, PMS, Hb, fecal calprotectin, and CRP values at 6 months of therapy did not have a statistically significant result (all *p* > 0.07).

At T12, a positive trend between T6 VDZ trough levels and Hb was found, with an r coefficient of 0.36 (*p* = 0.03), 95% IC 0.03–0.62, and reported in Figure 1.

A statistically significant association between VDZ trough levels and clinical remission at T12 was found. The odds ratio is 1.02 with a *p*-value of 0.03 (Figure 2).

Furthermore, a ROC analysis was performed to identify the optimal threshold value of VDZ levels associated with clinical remission at T12; through levels greater than 43.1 µg/mL, we were able to predict remission one year after therapy, with a sensitivity of 81.8% and a specificity of 55.6% (Figure 3).

A statistically significant association between the VDZ trough levels at T6 calprotectin <250 μg/g at T12 was found. The odds ratio is 1.01 with a *p*-value of 0.04. Furthermore, a ROC analysis was performed to identify the optimal threshold value of VDZ levels at T6 associated with calprotectin <250 μg/g at T12; through levels greater than 45.2 µg/mL, we were able to predict remission one year after therapy, with a sensitivity of 79.8% and a specificity of 52.6%.

For the prediction of clinical remission at T12, a univariate logistic regression was initially performed; subsequently, the variables that were statistically significant were entered in a multivariate analysis; the results are reported in Appendix A.

## 4. Discussion

The aim of the study was to evaluate VDZ exposure in ulcerative colitis and Crohn’s disease patients at six months of therapy, correlating it with the therapeutic outcome.

There is some evidence of a better outcome for patients with higher VDZ levels in terms of the clinical response [32,33] as well as the treatment persistence [34]. Another interesting study correlated VDZ trough levels with some laboratory parameters (CRP and Hb) evaluated at the same time point, demonstrating that higher drug concentrations correspond to lower CRP values and higher Hb, according to the results of the present study [35]. In our study, a positive trend (r = 0.36) was found between VDZ trough levels at T6 and Hb at T12, demonstrating the relevance of the TDM in predicting the efficacy of the biological therapy: patients with higher serum concentrations of the drug after six months of treatment tend to have higher Hb concentrations after a further six months, which is compatible with a better clinical outcome.

Furthermore, we found an association between VDZ levels at 6 months and clinical remission at 12 months of therapy. In particular, patients in clinical remission showed higher VDZ concentrations; this could be due to better inflammation management with VDZ, which is present in higher concentrations compared to patients without reaching the clinical response. Consequently, a ROC analysis was performed in order to suggest a possible VDZ cut-off value able to predict clinical remission one year after starting therapy; a value of 43.1 μg/mL was identified, with a sensitivity of 81.8% and a specificity of 55.6%. The association was confirmed with calprotectin <250 μg/g at T12, a surrogate marker of mucosal healing.

Finally, a logistic regression analysis was performed in order to evaluate which demographic, pharmacologic, and clinical variables were able to predict the clinical remission at twelve months of therapy. No factor was retained in the final multivariate regression model, probably due to the small number of patients recruited in this study, but further studies in larger cohorts of patients are ongoing in order to confirm or not confirm these preliminary results. 

In fact, the main limitation of the present study includes the small sample size and also the lack of an endoscopic examination obtained during the follow-up period that could more accurately describe the disease activity of the patients and the lack of a proactive intervention based on the results obtained from the TDM. Consequently, patients receiving VDZ every 4 weeks in the first 6 months of the follow-up were excluded from the study.

In conclusion, this is the first preliminary study reporting a possible VDZ serum cut-off value able to predict the clinical remission at twelve months of therapy, but further studies in larger cohorts of patients are needed to confirm our findings.

## Figures and Tables

**Figure 1 biomedicines-12-00824-f001:**
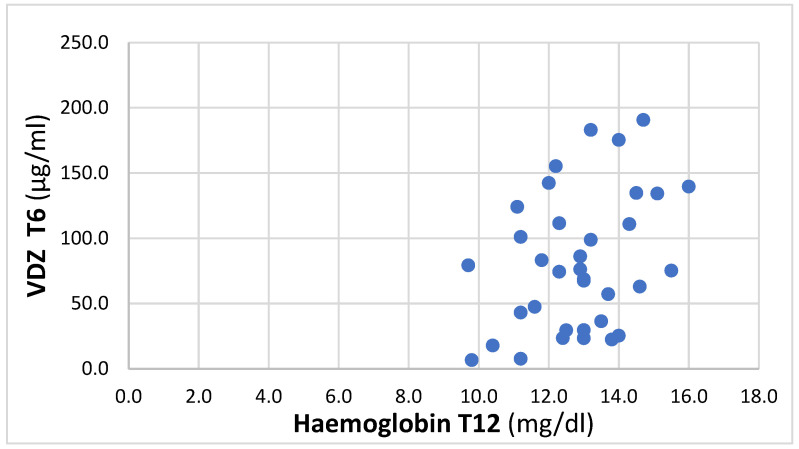
Correlation between VDZ at T6 and Hb at T12.

**Figure 2 biomedicines-12-00824-f002:**
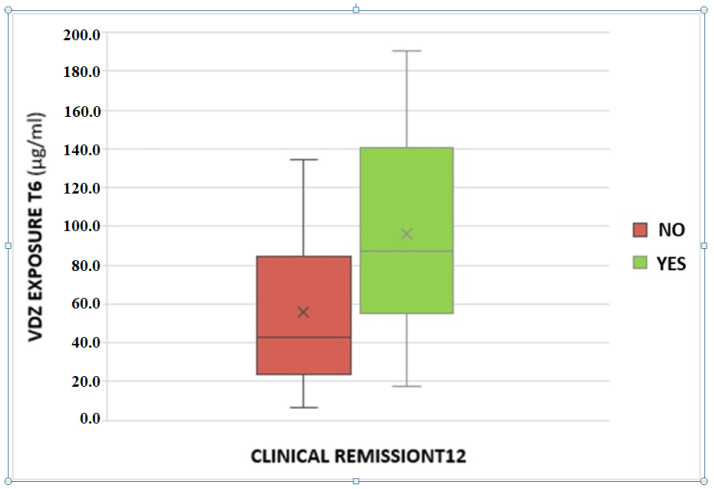
Association between serum vedolizumab levels at T6 and clinical remission at the T12.

**Figure 3 biomedicines-12-00824-f003:**
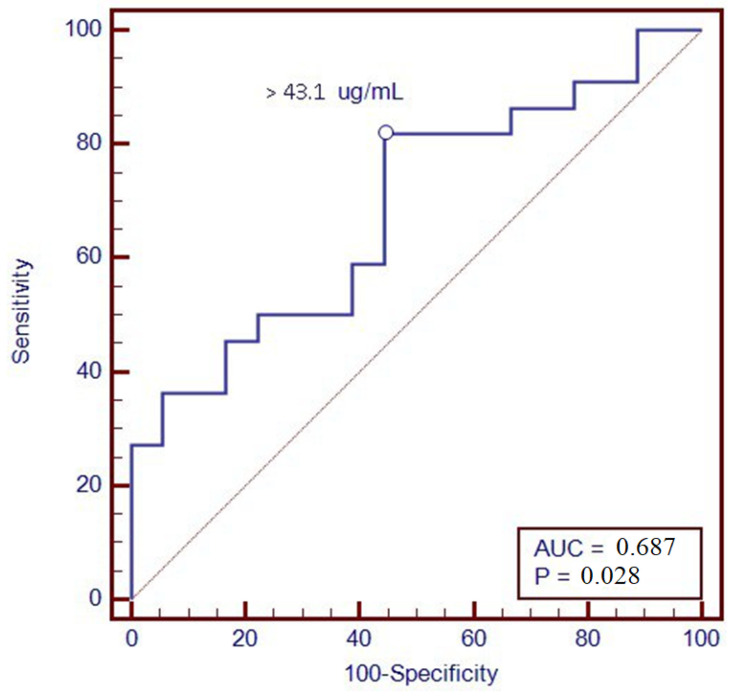
ROC curve of vedolizumab trough levels at T6 with the clinical remission at T12.

**Table 1 biomedicines-12-00824-t001:** Patients characteristics. SD = Standard Deviation.

Variables	
Male, *n* (%)	22 (55)
Age, median (inter quartile range)	53 (40.5–69.5)
BMI, mean (±SD)	23 ± 3.6
Active smokers, *n* (%)	5 (12.5)
Ex-smokers, *n* (%)	1 (2.5)
Ulcerative colitis, *n* (%)	24 (60)
Type of colitis	E1—proctosigmoiditis, *n* (%) 5 (20.8) E2—left colitis, *n* (%) 3 (12.5) E3—extensive colitis, *n* (%) 16 (66.7)
Crohn’s disease, *n* (%)	16 (40)
Type of Crohn’s disease	L1—ileal, *n* (%) 5 (31.3) L2—colic, *n* (%) 7 (43.7) L3—ileo-colic, *n* (%) 4 (25)
Years from diagnosis, median (inter quartile range)	16.5 (11.5–21.5)
5–aminosalicylic acid, *n* (%)	24 (60) (all Ulcerative colitis)
Topical corticosteroids, *n* (%)	1 (2.5) (all Ulcerative colitis)
Systemic corticosteroids, *n* (%)	16 (40) (10 Ulcerative colitis, 6 Crohn’s disease)
Topical and systemic corticosteroids, *n* (%)	7 (17.5) (all Ulcerative colitis)
Azathioprine, *n* (%)	4 (10) (all Ulcerative colitis)

**Table 2 biomedicines-12-00824-t002:** Clinical and hematochemical parameters at the different analyzed timings. SD = Standard Deviation.

Variables	HBI	PMS	CRP	Hemoglobin	Fecal Calprotectin	Vedolizumab Concentrations
T0 mean (±SD)/median (inter quartile range)	7.5 ± 4.4	4.9 ± 1.6	16 ± 40	12.2 ± 1.1	439 (244–1062)	
T6 mean (±SD)/median (inter quartile range)	3.5 ± 3.2	2.5 ± 2.5	15 ± 37.5	12.7 ± 1.7	196 (28–489)	79.3 µg/mL ± 51.3
T12 mean (±SD)/median (inter quartile range)	3.6 ± 3.1	1.5 ± 2	12 ± 34.3	12.8 ± 1.5	109 (15–312)	

CRP values statistically decreased from T0 to T12 (*p* = 0.04) but not from T6 to T12 (*p* = 0.19), calprotectin values statistically decreased both from T0 to T12 (*p* = 0.009) and from T0 to T6 (*p* = 0.01).

**Table 3 biomedicines-12-00824-t003:** Corticosteroid-free clinical response and remission during the follow-up.

	N/TOT (%)
Corticosteroid-free clinical response T6	31/40 (77.5)
Corticosteroid-free clinical remission T6	25/40 (62.5)
Corticosteroid-free clinical response T12	27/35 (77.1)
Corticosteroid-free clinical remission T12	22/35 (62.9)

## Data Availability

Data are available within the text. Patient data are available on request due to privacy and ethical restrictions.

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
