# Peer review of "Therapeutic Drug Monitoring as a Tool for the Clinical Outcome Prediction in Vedolizumab-Treated Patients: An Italian Pilot Study"

_biomedicines, 2024, doi:10.3390/biomedicines12040824_

Round 1
Reviewer 1 Report
Comments and Suggestions for Authors
Dear authors,
After reviewing this study, I have several concerns.
1. The introduction is too long and contains several non-essential information. Introduction needs to emphasize the recent protocols, guidelines, or conventions and the unmet medical needs derived from that. Clearance of velidozumab cannot help the reader to understand the unmet medical need of velidozumab application.
2. Please unify the presentation of the table. In table 1, 2, 4, the authors shows full line of the column. However, in figure 3, the authors use three-line table. Please unify them.
3. The authors correlated velidozumab concentration at T6 time point and hemoglobin content at T12 time point but such correlation is very uncommon. Please explain why the authors perform such correlation.
4. Continuous from the previous concern, the authors identify such correlation but do not attend to explain the underlying mechanism. Please provide your discussion.
Comments on the Quality of English LanguageI cannot identify the grammar errors in the presentation. However, the wording of this study is hard to understand.
Author Response
R: Thank you for your revisions
- The introduction is too long and contains several non-essential information. Introduction needs to emphasize the recent protocols, guidelines, or conventions and the unmet medical needs derived from that. Clearance of velidozumab cannot help the reader to understand the unmet medical need of velidozumab application.
R: Thank you for your comment, but the journal requires at least 4000 words. We tried to highlight the most important concepts.
- Please unify the presentation of the table. In table 1, 2, 4, the authors shows full line of the column. However, in figure 3, the authors use three-line table. Please unify them.
R: Thank you for your comment. We unified the presentation of the tables, as required.
- The authors correlated velidozumab concentration at T6 time point and hemoglobin content at T12 time point but such correlation is very uncommon. Please explain why the authors perform such correlation.
R: thank you for question. We decided to investigate if some factors could be predictors of variables at 12 months of therapy. Consequently, we would like to know if vedolizumab concentrations at 6 months are able to predict factors at 12 months of follow up.
- Continuous from the previous concern, the authors identify such correlation but do not attend to explain the underlying mechanism. Please provide your discussion.
R: thank you, we modified the discussion according to your recommendation.
I cannot identify the grammar errors in the presentation. However, the wording of this study is hard to understand
R: thank you for your comment. English has been revised, according to your recommendation, making it more simple to understand.
Reviewer 2 Report
Comments and Suggestions for Authors
Although, as the Authors wrote, “this is the first study reporting a possible VDZ serum cut-off value able to predict the clinical remission at twelve months of therapy”, there are many limitations, which preclude the manuscript publication in the present from. Some limitations were mentioned by the Authors, but it is not enough to be mentioned, in order to make the research more accurate. Normally, I would decide on “Rejection”, but maybe the Authors could improve their manuscript, accordingly.
A. Major comments:
- Too few patients
- No separate analyses for UC and CD
- No mucosal healing assessed
- No factor as predictor of clinical remission at T 12 in multivariate regression analysis
- r coefficient too low
- ROC too low specificity
- Appreciation of just clinical response/remission
- No real significance of the results (in real world), which are too flimsy
B. My comments in order of their appearance, while reading the manuscript (which would be the same for potential readers):
1. Title: After reading the whole manuscript, I do not see why TDM represents” a “diagnostic tool”; it is not “diagnostic”.
2. Abstract:
a. Please revise “more targeted and more effective target therapy”, as it is redundant/target is repeating
b. Please mention when the study was conducted. Please mention how many had UC and CD, and whether there was a difference between them.
c. Clinical remission is not enough.
d. I found out later that CRP and fecal calprotectin were performed, but this was not mentioned. Please add.
e. “In conclusion, this is the first study reporting a possible VDZ serum cut-off value able to predict the clinical remission at twelve months of therapy.”. Here and in the whole manuscript, sometimes it is written “at”, sometimes “after”. Please decide and unify.
3. Introduction:
a. Please use more recent references. We are in 2024.
b. Line 39: “increasingly frequent worldwide”? please revise.
c. Also, for increasing incidence/prevalence, you could use:
“Wang R, Li Z, Liu S, Zhang D. Global, regional and national burden of inflammatory bowel disease in 204 countries and territories from 1990 to 2019: a systematic analysis based on the Global Burden of Disease Study 2019. BMJ Open. 2023 Mar 28;13(3):e065186. doi: 10.1136/bmjopen-2022-065186. PMID: 36977543”
“GBD 2017 Inflammatory Bowel Disease Collaborators. The global, regional, and national burden of inflammatory bowel disease in 195 countries and territories, 1990–2017: a systematic analysis for the Global Burden of Disease Study 2017. Lancet Gastroenterol Hepatol. 2020;5:17–30”
“Kaplan GG, Windsor JW. The four epidemiological stages in the global evolution of inflammatory bowel disease. Nat Rev Gastroenterol Hepatol. 2021 Jan;18(1):56-66.”
d. Reference 2 – it is not from 2019, but from 2020. Please correct. And it is just for Crohn’s; please insert for UC as well, from 2022 (Raine T, et al). In any case, many more were approved after 2020, respectively 2022. You could use more recent updated wonderful manuscripts, reviewing all these new molecules.
e. Reference 3 – to be replaced as well (e.g. Annese V, Annese M. Precision Medicine in Inflammatory Bowel Disease. Diagnostics (Basel). 2023 Aug 29;13(17):2797.)
f. Please update all references.
g. Why only clinical remission? Nowadays, the accepted target is represented by mucosal healing. Some examples, that had been published even years ago:
“Al-Bawardy B, Ramos GP, Willrich MAV, Jenkins SM, Park SH, Aniwan S, Schoenoff SA, Bruining DH, Papadakis KA, Raffals L, Tremaine WJ, Loftus EV. Vedolizumab Drug Level Correlation With Clinical Remission, Biomarker Normalization, and Mucosal Healing in Inflammatory Bowel Disease. Inflamm Bowel Dis. 2019 Feb 21;25(3):580-586.” AND MANY PATIENTS WERE INCLUDED!
“Plevris N, Jenkinson PW, Chuah CS, Lyons M, Merchant LM, Pattenden RJ, Arnott ID, Jones GR, Lees CW. Association of trough vedolizumab levels with clinical, biological and endoscopic outcomes during maintenance therapy in inflammatory bowel disease. Frontline Gastroenterol. 2019 Jul 3;11(2):117-123.” AND MANY PATIENTS WERE INCLUDED!
“Hüttemann E, Muzalyova A, Gröhl K, Nagl S, Fleischmann C, Ebigbo A, Classen J, Wanzl J, Prinz F, Mayr P, Schnoy E. Efficacy and Safety of Vedolizumab in Patients with Inflammatory Bowel Disease in Association with Vedolizumab Drug Levels. J Clin Med. 2023 Dec 27;13(1):140.” AND MANY PATIENTS WERE INCLUDED!
Etc….
f. Reference 14 is not complete. Please correct.
g. There are many very recent references to use about TDM in IBD. Sentences from line 88 to 97 could be deleted. Not related to the topic.
h. Please again, replace ref. [19] and [20], they are ancient (2009 and 2011).
i. Same for references [21] and [22], they are ancient (2012 and 2014). Nowadays, an IFX level of at least 5 microg/ml is required in the guidelines.
j. Aim, by the end of Introduction: it was mentioned “at twelve months”, but in the Abstract it was also written “after” twelve months. Please clarify. And, as I mentioned before, clinical response/remission is not enough nowadays. Just look at the reference you inserted [23] – STRIDE II. Also, what about CRP? Fecal calprotectin? Endoscopic assessment? Data must be completed. I saw later that CRP was considered as well as fecal calprotectin, but this was not mentioned before. Some statistics could be done then.
4. Materials and Methods
a. Study Design:
a1. It appears there were only 2 centers, not “multi”.
a2. Line 127: Now, it is written again – “after” 12 months, not “at”. It is not the same, obviously. Please revise and clarify.
a3. It appears that fecal calprotectin test was performed. Then, please do not mention only “clinical remission”. Very confusing. Why were serum inflammatory biological markers not measured? Why just “Hb”? I saw later in Results that, indeed, CRP was measured. Then, please mention it and do something with that.
b. Statistical analysis: Lines 162- 164: “In the multivariate analysis the statistically significant variables were included in the univariate analysis (enter regression).” It should be vice versa.
5. Results
a. There is only a paragraph: “3.1.Patients Characteristics”, there is no 3.2. Then, why bothering inserting 3.1?
b. Table 1 could be better organized.
b1. Also, the name of the disease is misspelled: “Chron’s” instead of “Crohn’s”. I wonder, did the Authors double check their manuscript?
b2. Please insert behaviour for CD and consider it in the regression analysis.
b3. Please write in details therapy at T0 separately, for UC and CD. And what type of immunomodulators, please? AZA? MTX?
c. Please reorganize Table 2.
c1. Please put T0, T6 and T 12 horizontally, at the beginning of the table (first raw), so that data for each parameter becomes easily to be followed (HBI, PMS etc), horizontally.
c2. From Table 2, it appears that C-RP was also determined (and it was never defined before as C- reactive protein), which was not at all mentioned before. I just wrote above why was it not measured. This is crucial, as if the readers find out it just in the Results, it is not really helpful. The abstract mentions only “clinical” data. Please assess this also in Material and Methods.
d. Please perform statistics regarding levels of C-RP and fecal calprotectin at T0, T6 and T12.
e. “At T6, correlations between the trough levels of VDZ with HBI, PMS, Hb, faecal calprotectin values at 6 months of therapy did not result statistically significant (all p > 0.07).”. Please mention also about C-RP.
f. Please perform analyses separately for UC and CD.
g. Line 187: “r coefficient of 0.36” is not good enough. And, why did the Authors choose the Hb value?
h. Line 197: “specificity of 55.6 %” is not good enough.
i. What about VDZ levels at T 12?
j. UC was not included in Table 4.
k. Also, it seems that no factor was statistically significant in multivariate regression analysis. This is very poignant.
6. Discussion
a. The authors should start with their own findings (not just theory), and relate them to the existing studies. Those sentences could be nicely introduced in Introduction”, with appropriate recent references.
b. Lines 241-242: “laboratory parameters (C-reactive protein, CRP, and Hb)” – please delete CRP as it refers to C-reactive protein.
c. There is no proper “Discussion” here, just some studies from the literature commented. In fact, the results are so flimsy, that what to comment on? However, I would suggest to focus on CRP and fecal calprotectin, to make the best of it.
7. References: as I mentioned, some of them are ancient. Please update.
At least 7 references by the Authors, out of 33 (21%).
8. I read the iThenticate report in detail and it looks fine.
Comments on the Quality of English LanguageCould be better (e.g. missing commas, non-agreement verb-noun, etc.)
Author Response
Although, as the Authors wrote, “this is the first study reporting a possible VDZ serum cut-off value able to predict the clinical remission at twelve months of therapy”, there are many limitations, which preclude the manuscript publication in the present from. Some limitations were mentioned by the Authors, but it is not enough to be mentioned, in order to make the research more accurate. Normally, I would decide on “Rejection”, but maybe the Authors could improve their manuscript, accordingly.
- Major comments:
- Too few patients
- No separate analyses for UC and CD
- No mucosal healing assessed
- No factor as predictor of clinical remission at T 12 in multivariate regression analysis
- r coefficient too low
- ROC too low specificity
- Appreciation of just clinical response/remission
- No real significance of the results (in real world), which are too flimsy
R: thank you for your right comments. We agree with you, the size is small, as reported in the limitations. We can suggest in the manuscript it is a pilot explorative study to be confirmed in further studies, as we are performing. In addition, we think the reduced statistical power is linked to the small number of patients, but this could be the starting point for larger analyses we are performing. Basically, no division was performed between UC and CD, according to clinician recommendations, in order to have a larger sample size and, moreover, Vedolizumab has similar efficacies both for UC and CD. In addition, it is correct to consider another outcome and not only the clinical outcome, but we choose an important clinical outcome as the steroids-free remission, as for example reported in other studies (D’Haens, Inflamm. Bowel Dis. 2012; Penna, BMC Gastroenterol. 2020). Another important issue to highlight is that some hospitals could decide to perform or not the endoscopy within 1 year in patients with a good response for biological drugs, particularly in Crohn’s disease, if the drug is not used as first-line advanced therapy and if they intend to continue the drug, as one of the two centres. Consequently, due to these suggestions we decided to add the investigation of a calprotectin < 250, a surrogate biomarker of mucosal healing.
- My comments in order of their appearance, while reading the manuscript (which would be the same for potential readers):
- Title: After reading the whole manuscript, I do not see why TDM represents” a “diagnostic tool”; it is not “diagnostic”.
R: thanks for your comment. We agree with you and we removed the term “diagnostic”.
- Abstract:
R: thank you for revision. We tried to modify the abstract according to your recommendations, but we have to consider that the journal requires only 200 words for the abstract.
- Please revise “more targeted and more effective target therapy”, as it is redundant/target is repeating
R: thank you. We changed the sentence, according to your comment.
- Please mention when the study was conducted. Please mention how many had UC and CD, and whether there was a difference between them.
R: thank you. We added the information of how many UC and CD in the manuscript, since it was written only in table 1.
- Clinical remission is not enough.
R: we are sorry, but the abstract limit dos not lead us to add this data.
- I found out later that CRP and fecal calprotectin were performed, but this was not mentioned. Please add.
R: these data are added in the manuscript materials and methods, according to your revision.
- “In conclusion, this is the first study reporting a possible VDZ serum cut-off value able to predict the clinical remission at twelve months of therapy.”. Here and in the whole manuscript, sometimes it is written “at”, sometimes “after”. Please decide and unify.
R: thank you for your right comment. We revised all the abstract and manuscript.
- Introduction:
- Please use more recent references. We are in 2024.
R: thank you. We changed the references according to your suggestions.
- Line 39: “increasingly frequent worldwide”? please revise.
R: thank you. We revise it.
- Also, for increasing incidence/prevalence, you could use:
“Wang R, Li Z, Liu S, Zhang D. Global, regional and national burden of inflammatory bowel disease in 204 countries and territories from 1990 to 2019: a systematic analysis based on the Global Burden of Disease Study 2019. BMJ Open. 2023 Mar 28;13(3):e065186. doi: 10.1136/bmjopen-2022-065186. PMID: 36977543”
“GBD 2017 Inflammatory Bowel Disease Collaborators. The global, regional, and national burden of inflammatory bowel disease in 195 countries and territories, 1990–2017: a systematic analysis for the Global Burden of Disease Study 2017. Lancet Gastroenterol Hepatol. 2020;5:17–30”
“Kaplan GG, Windsor JW. The four epidemiological stages in the global evolution of inflammatory bowel disease. Nat Rev Gastroenterol Hepatol. 2021 Jan;18(1):56-66.”
- Reference 2 – it is not from 2019, but from 2020. Please correct. And it is just for Crohn’s; please insert for UC as well, from 2022 (Raine T, et al). In any case, many more were approved after 2020, respectively 2022. You could use more recent updated wonderful manuscripts, reviewing all these new molecules.
- Reference 3 – to be replaced as well (e.g. Annese V, Annese M. Precision Medicine in Inflammatory Bowel Disease. Diagnostics (Basel). 2023 Aug 29;13(17):2797.)
- Please update all references.
R: thank you. We updated all the references.
- Why only clinical remission? Nowadays, the accepted target is represented by mucosal healing. Some examples, that had been published even years ago:
“Al-Bawardy B, Ramos GP, Willrich MAV, Jenkins SM, Park SH, Aniwan S, Schoenoff SA, Bruining DH, Papadakis KA, Raffals L, Tremaine WJ, Loftus EV. Vedolizumab Drug Level Correlation With Clinical Remission, Biomarker Normalization, and Mucosal Healing in Inflammatory Bowel Disease. Inflamm Bowel Dis. 2019 Feb 21;25(3):580-586.” AND MANY PATIENTS WERE INCLUDED!
“Plevris N, Jenkinson PW, Chuah CS, Lyons M, Merchant LM, Pattenden RJ, Arnott ID, Jones GR, Lees CW. Association of trough vedolizumab levels with clinical, biological and endoscopic outcomes during maintenance therapy in inflammatory bowel disease. Frontline Gastroenterol. 2019 Jul 3;11(2):117-123.” AND MANY PATIENTS WERE INCLUDED!
“Hüttemann E, Muzalyova A, Gröhl K, Nagl S, Fleischmann C, Ebigbo A, Classen J, Wanzl J, Prinz F, Mayr P, Schnoy E. Efficacy and Safety of Vedolizumab in Patients with Inflammatory Bowel Disease in Association with Vedolizumab Drug Levels. J Clin Med. 2023 Dec 27;13(1):140.” AND MANY PATIENTS WERE INCLUDED!
Etc….
R.: Thank you. It is correct to consider another outcome and not only the clinical outcome, but we choose an important clinical outcome as the remission steroids free, as for example reported in other studies (D’Haens, Inflamm. Bowel Dis. 2012; Penna, BMC Gastroenterol. 2020). Another important issue to highlight is that some hospitals could decide to perform or not the endoscopy in patients with a good response for biological drugs, as one of the two centres. Consequently, due to these suggestions we decided to add the investigation of a calprotectin < 250 μg/g at T12.
- Reference 14 is not complete. Please correct.
R: thank you. The reference was corrected.
- There are many very recent references to use about TDM in IBD. Sentences from line 88 to 97 could be deleted. Not related to the topic.
R: thank you. We decided to maintain this part because we believe it could be useful for the reader in order to have a global idea of Vedolizumab.
- Please again, replace ref. [19] and [20], they are ancient (2009 and 2011).
R: thank you. The reference was replaced.
- Same for references [21] and [22], they are ancient (2012 and 2014). Nowadays, an IFX level of at least 5 microg/ml is required in the guidelines.
R: thank you. The reference was replaced.
- Aim, by the end of Introduction: it was mentioned “at twelve months”, but in the Abstract it was also written “after” twelve months. Please clarify. And, as I mentioned before, clinical response/remission is not enough nowadays. Just look at the reference you inserted [23] – STRIDE II. Also, what about CRP? Fecal calprotectin? Endoscopic assessment? Data must be completed. I saw later that CRP was considered as well as fecal calprotectin, but this was not mentioned before. Some statistics could be done then.
R.: Thank you. As reported above, it is correct to consider another outcome and not only the clinical outcome, but we choose an important clinical outcome as the remission steroids free, as for example reported in other studies (D’Haens, Inflamm. Bowel Dis. 2012; Penna, BMC Gastroenterol. 2020). Another important issue to highlight is that some hospitals could decide to perform or not the endoscopy in patients with a good response for biological drugs, as one of the two centres. Consequently, due to these suggestions we decided to add the investigation of a calprotectin < 250 μg/g at T12..
- Materials and Methods
- Study Design:
a1. It appears there were only 2 centers, not “multi”.
R: we agree with you and changed it, also in the title.
a2. Line 127: Now, it is written again – “after” 12 months, not “at”. It is not the same, obviously. Please revise and clarify.
R: we revised it in the entire manuscript, as suggested.
a3. It appears that fecal calprotectin test was performed. Then, please do not mention only “clinical remission”. Very confusing. Why were serum inflammatory biological markers not measured? Why just “Hb”? I saw later in Results that, indeed, CRP was measured. Then, please mention it and do something with that.
R: Thank you for your comment. Due to these suggestions we decided to add the investigation of a calprotectin < 250 μg/g at T12..
- Statistical analysis: Lines 162- 164: “In the multivariate analysis the statistically significant variables were included in the univariate analysis (enter regression).” It should be vice versa.
R: sorry for the mistake. We revise it, thank you!
- Results
- There is only a paragraph: “3.1.Patients Characteristics”, there is no 3.2. Then, why bothering inserting 3.1?
R: thank you. We deleted the 3.1 paragraph.
- Table 1 could be better organized.
R: thank you, we revised it.
b1. Also, the name of the disease is misspelled: “Chron’s” instead of “Crohn’s”. I wonder, did the Authors double check their manuscript?
R: sorry for the mistakes. We corrected it, according to your revision.
b2. Please insert behaviour for CD and consider it in the regression analysis.
R: We did not include this sub-analysis to not reduce the power of the study (too few patients in three subgroups of one of the two diseases included).
b3. Please write in details therapy at T0 separately, for UC and CD. And what type of immunomodulators, please? AZA? MTX?
R: We added the details.
- Please reorganize Table 2.
R: thank you. It was reorganized, according to your suggestions.
c1. Please put T0, T6 and T 12 horizontally, at the beginning of the table (first raw), so that data for each parameter becomes easily to be followed (HBI, PMS etc), horizontally.
R: thank you for your suggestion. We revise it.
c2. From Table 2, it appears that C-RP was also determined (and it was never defined before as C- reactive protein), which was not at all mentioned before. I just wrote above why was it not measured. This is crucial, as if the readers find out it just in the Results, it is not really helpful. The abstract mentions only “clinical” data. Please assess this also in Material and Methods.
R: thank you. We agree with you and correct this part by adding it in materials and methods.
- Please perform statistics regarding levels of C-RP and fecal calprotectin at T0, T6 and T12.
R: We added the statistical comparisons.
- “At T6, correlations between the trough levels of VDZ with HBI, PMS, Hb, faecal calprotectin values at 6 months of therapy did not result statistically significant (all p > 0.07).”. Please mention also about C-RP.
R: We added the correlation with C-RP.
- Please perform analyses separately for UC and CD.
R: thank you for your right comment, but according to the clinicians we decided to perform all the analysis regarding the two different pathologies together due to the small sample size and considering that Vedolizumab efficacy in these two diseases is similar.
- Line 187: “r coefficient of 0.36” is not good enough. And, why did the Authors choose the Hb value?
R: thank you. As reported above this a pilot explorative study laying the foundations for further investigations which are ongoing. Thus the statistical power is not so high.
- Line 197: “specificity of 55.6 %” is not good enough.
R: thank you. As reported above this a pilot explorative study laying the foundations for further investigations which are ongoing. Thus the statistical power is not so high. We highlighted this point in the manuscript.
- What about VDZ levels at T 12?
R: we decided to no evaluate vedolizumab T12 concentrations since we are searching for predictors of clinical remission at 12 months of therapy; thus potential predictors have to be evaluated before the event.
- UC was not included in Table 4.
R: We added UC:
- Also, it seems that no factor was statistically significant in multivariate regression analysis. This is very poignant.
R: thanks for your kind analysis, but, as suggested before, this is a pilot explorative study focusing on a small number of patients. Thus, this could be the reason why no variables resulted statistically significant in the regression model.
- Discussion
- The authors should start with their own findings (not just theory), and relate them to the existing studies. Those sentences could be nicely introduced in Introduction”, with appropriate recent references.
R: thank you, we added these parts in the introduction, as recommended.
- Lines 241-242: “laboratory parameters (C-reactive protein, CRP, and Hb)” – please delete CRP as it refers to C-reactive protein
R: sorry for the mistake
- There is no proper “Discussion” here, just some studies from the literature commented. In fact, the results are so flimsy, that what to comment on? However, I would suggest to focus on CRP and fecal calprotectin, to make the best of it.
R: thank you, we changed the discussion according to your suggestions
- References: as I mentioned, some of them are ancient. Please update.
At least 7 references by the Authors, out of 33 (21%).
R: thank you for your comment, we changed the references
- I read the iThenticate report in detail and it looks fine.
R: thank you
Reviewer 3 Report
Comments and Suggestions for Authors
The Authors applied the therapeutic drug monitoring in the clinical outcome prediction in vedolizumab.
The results might be considered as interesting, but some issues should be improved:
1. What were the inclusion criteria to the study?
2. Did Authors take into consideration also the gender of the patient? In my opinion it should be also checked
3. I doubt that correlation r=0.36 is a good correlation. It is positive - I agree. It may show that is a positive trend.
4. I would put the VDZ levels in the table. It is hard to find them in a main text.
5. Some editorial issues: Authors should remove the vertical and horizontal lines inside the tables. It is hard to read them. Besides, I would use SD instead of standard deviation. In table 4 are there two sections concerning either univariate analyis and multivariate analysis. The second one is rather empty. We may find the parameters conerning PMS T6, C-RP T6, Haemoglobin T0, T6 and VDZ concentrations T6. Maybe these parameters should be presented in a separate table with uni- and multivariate analysis. The other option in the transfer this table to supplementary information. In line 147 there is Ctrough. It should be Ctrough.
Author Response
The Authors applied the therapeutic drug monitoring in the clinical outcome prediction in vedolizumab.
The results might be considered as interesting, but some issues should be improved:
R: thank you for your revision
- What were the inclusion criteria to the study?
R: thank you for your comment, we added the inclusion criteria in the materials and methods
- Did Authors take into consideration also the gender of the patient? In my opinion it should be also checked
R: thank you, we investigated gender role, according to your recommendation (see Table S1, Male sex).
- I doubt that correlation r=0.36 is a good correlation. It is positive - I agree. It may show that is a positive trend.
R: we agree with you, we changed the sentence.
- I would put the VDZ levels in the table. It is hard to find them in a main text.
R: thank you for your comment, we add this information in the table
- Some editorial issues: Authors should remove the vertical and horizontal lines inside the tables. It is hard to read them. Besides, I would use SD instead of standard deviation. In table 4 are there two sections concerning either univariate analyis and multivariate analysis. The second one is rather empty. We may find the parameters conerning PMS T6, C-RP T6, Haemoglobin T0, T6 and VDZ concentrations T6. Maybe these parameters should be presented in a separate table with uni- and multivariate analysis. The other option in the transfer this table to supplementary information. In line 147 there is Ctrough. It should be Ctrough.
R: thank you for your comments, we changed the sentences, according to your suggestions.
Reviewer 4 Report
Comments and Suggestions for Authors
Dear authors
it is an intresting study. I would like please to add/comment more extensively:
1. the registration number to clinicalTrials.gov - it is a prospective study
2. was there a power analysis? Forty patients - two diseases and sup-types due to the parts of the gut involved make the number of patient included rather small
3. this [the above] would be the cause of no results found in logistic regression analysis performed
Author Response
Dear authors
it is an intresting study. I would like please to add/comment more extensively:
- the registration number to clinicalTrials.gov - it is a prospective study
R: Thank you. It is not a trial, we only have the ethical committee approval, as reported in manuscript.
- was there a power analysis? Forty patients - two diseases and sup-types due to the parts of the gut involved make the number of patient included rather small
R: thank you, we decided to change with “pilot explorative study” since a small number of patients was analyzed, as you suggested. Other analysis are underway in order to better define these data.
- this [the above] would be the cause of no results found in logistic regression analysis performed
R: we agree with you, this could be the reason. We highlight this limitation in the manuscript.
Round 2
Reviewer 2 Report
Comments and Suggestions for Authors
The manuscript was improved.
Suggestions/comments for minor revision:
1. Please insert reference after the paragraph (line 160).
2. I kindly asked Table 2 to be modified, having columns with T0, T6 and T12, for each parameter. “Please put T0, T6 and T 12 horizontally, at the beginning of the table (first raw), so that data for each parameter becomes easily to be followed (HBI, PMS etc), horizontally.”
3. Also, I meant that since CRP and fecal calprotectin were assessed and calprotectin was significantly correlated with VDZ levels, then it was more than clinical remission. As we all know, fecal calprotectin is considered as a surrogate marker for mucosal healing. This is why, the Abstract should mention more than “clinical remission”, as the manuscript shows more aspects and has MUCH BETTER VALUE. The authors should value more their manuscript.
4. I wonder why previous good manuscripts like those I previously mentioned, were not included. They are truly good references. None involves me or anyone I know.
“Al-Bawardy B, Ramos GP, Willrich MAV, Jenkins SM, Park SH, Aniwan S, Schoenoff SA, Bruining DH, Papadakis KA, Raffals L, Tremaine WJ, Loftus EV. Vedolizumab Drug Level Correlation With Clinical Remission, Biomarker Normalization, and Mucosal Healing in Inflammatory Bowel Disease. Inflamm Bowel Dis. 2019 Feb 21;25(3):580-586.” AND MANY PATIENTS WERE INCLUDED!
“Plevris N, Jenkinson PW, Chuah CS, Lyons M, Merchant LM, Pattenden RJ, Arnott ID, Jones GR, Lees CW. Association of trough vedolizumab levels with clinical, biological and endoscopic outcomes during maintenance therapy in inflammatory bowel disease. Frontline Gastroenterol. 2019 Jul 3;11(2):117-123.” AND MANY PATIENTS WERE INCLUDED!
“Hüttemann E, Muzalyova A, Gröhl K, Nagl S, Fleischmann C, Ebigbo A, Classen J, Wanzl J, Prinz F, Mayr P, Schnoy E. Efficacy and Safety of Vedolizumab in Patients with Inflammatory Bowel Disease in Association with Vedolizumab Drug Levels. J Clin Med. 2023 Dec 27;13(1):140.” AND MANY PATIENTS WERE INCLUDED!”
Thank you
Comments on the Quality of English LanguageGenerally good; there are some typos and misspelled words that require correction.
Author Response
- Please insert reference after the paragraph (line 160).
R: Thank you for your revision. Reference has been insert after the paragraph.
- I kindly asked Table 2 to be modified, having columns with T0, T6 and T12, for each parameter. “Please put T0, T6 and T 12 horizontally, at the beginning of the table (first raw), so that data for each parameter becomes easily to be followed (HBI, PMS etc), horizontally.”
R: Thank you for your comment. Table 2 has been re-organized, according to your recommedations.
- Also, I meant that since CRP and fecal calprotectin were assessed and calprotectin was significantly correlated with VDZ levels, then it was more than clinical remission. As we all know, fecal calprotectin is considered as a surrogate marker for mucosal healing. This is why, the Abstract should mention more than “clinical remission”, as the manuscript shows more aspects and has MUCH BETTER VALUE. The authors should value more their manuscript.
R: Thank you for your comment. The abstract has been revised.
- I wonder why previous good manuscripts like those I previously mentioned, were not included. They are truly good references. None involves me or anyone I know.
“Al-Bawardy B, Ramos GP, Willrich MAV, Jenkins SM, Park SH, Aniwan S, Schoenoff SA, Bruining DH, Papadakis KA, Raffals L, Tremaine WJ, Loftus EV. Vedolizumab Drug Level Correlation With Clinical Remission, Biomarker Normalization, and Mucosal Healing in Inflammatory Bowel Disease. Inflamm Bowel Dis. 2019 Feb 21;25(3):580-586.” AND MANY PATIENTS WERE INCLUDED!
“Plevris N, Jenkinson PW, Chuah CS, Lyons M, Merchant LM, Pattenden RJ, Arnott ID, Jones GR, Lees CW. Association of trough vedolizumab levels with clinical, biological and endoscopic outcomes during maintenance therapy in inflammatory bowel disease. Frontline Gastroenterol. 2019 Jul 3;11(2):117-123.” AND MANY PATIENTS WERE INCLUDED!
“Hüttemann E, Muzalyova A, Gröhl K, Nagl S, Fleischmann C, Ebigbo A, Classen J, Wanzl J, Prinz F, Mayr P, Schnoy E. Efficacy and Safety of Vedolizumab in Patients with Inflammatory Bowel Disease in Association with Vedolizumab Drug Levels. J Clin Med. 2023 Dec 27;13(1):140.” AND MANY PATIENTS WERE INCLUDED!”
R: Thank you for your revision. References were added to the manuscript.
Reviewer 3 Report
Comments and Suggestions for Authors
The alternations improved the quality of the manuscript.
Author Response
Thank you for your revision!